# Redox dynamics and surface structures of an active palladium catalyst during methane oxidation

Shengnan Yue [1,2,10], C. S. Praveen [3,10], Alexander Klyushin [4,10], Alexey Fedorov [5], Masahiro Hashimoto[6], Qian Li[1,2], Travis Jones [7] ✉, Panpan Liu[1,2], Wenqian Yu[1,2], Marc-Georg Willinger [8,9] & Xing Huang [1,2,8] ✉

Catalysts based on palladium are among the most effective in the complete oxidation of methane. Despite extensive studies and notable advances, the nature of their catalytically active species and conceivable structural dynamics remains only partially understood. Here, we combine *operando* transmission electron microscopy (TEM) with near-ambient pressure X-ray photoelectron spectroscopy (NAP-XPS) and density functional theory (DFT) calculations to investigate the active state and catalytic function of Pd nanoparticles (NPs) under methane oxidation conditions. We show that the particle size, phase composition and dynamics respond appreciably to changes in the gas-phase chemical potential. In combination with mass spectrometry (MS) conducted simultaneously with in situ observations, we uncover that the catalytically active state exhibits phase coexistence and oscillatory phase transitions between Pd and PdO. Aided by DFT calculations, we provide a rationale for the observed redox dynamics and demonstrate that the emergence of catalytic activity is related to the dynamic interplay between coexisting phases, with the resulting strained PdO having more favorable energetics for methane oxidation.

Natural gas engines have become a promising alternative to traditional petrol and diesel engines owing to the high energy density of $CH_4$ and reduced $NO_x$ and $CO_2$ emissions[1–3]. However, the lean-burn operation of natural gas engines typically leads to incomplete oxidation of $CH_4$ and yields unburned $CH_4$ in the exhaust[2,4]. This is unwanted since $CH_4$ is a more potent greenhouse gas than $CO_2$[5–7]. To minimize the $CH_4$ emission, catalytic conversion of unburned $CH_4$ to $CO_2$ and $H_2O$ is required. Among various materials, Pd-based catalysts have been recognized as the most effective in the complete oxidation of $CH_4$[1,3,5,8,9]. However, while significant research efforts have been devoted to this catalytic system, our understanding of the working state of Pd catalysts is still insufficient[10–13]. In particular, there is debate over the nature of the active surface. Some reports suggest that metallic Pd is more active than PdO in the complete oxidation of methane[14–17], however, most recent studies attribute the catalytic activity to $PdO_x$ or the presence of a metal/oxide interface[4,18–21]. These divergent conclusions may be linked to a dynamic coexistence of Pd and PdO under reaction conditions, making the assignment of distinct active structures and the establishment of structure–activity relationships challenging.

[1]College of Chemistry, Fuzhou University, Fuzhou, China. [2]Qingyuan Innovation Laboratory, Quanzhou, China. [3]International School of Photonics, Cochin University of Science and Technology, Cochin, Kerala, India. [4]MAX IV Laboratory, Lund University, Lund, Sweden. [5]Department of Mechanical and Process Engineering, ETH Zurich, Zurich, Switzerland. [6]JEOL (EUROPE) SAS, allée de Giverny, Croissy-sur-Seine, France. [7]Theoretical Division, Los Alamos National Laboratory, Los Alamos, NM, USA. [8]Scientific Center for Optical and Electron Microscopy, ETH Zurich, Zurich, Switzerland. [9]Department of Chemistry, Technical University of Munich, Garching, Germany. [10]These authors contributed equally: Shengnan Yue, C. S. Praveen, Alexander Klyushin. ✉e-mail: tejones@lanl.gov; xinghuang@fzu.edu.cn

Recent advances in the application of in situ and *operando* techniques in heterogeneous catalysis have enabled detailed insights into the working state of various catalysts[22–25]. Among these techniques, in situ transmission electron microscopy (TEM) is a particularly powerful tool for studying the atomic structure and dynamic behavior of materials as it offers real-time and real-space imaging of catalysts with high temporal and spatial resolution under external stimuli[26–33]. In particular, the combined use of online mass spectrometry (MS) with in situ TEM has demonstrated great potential in improving our understanding of the structure-performance relationships in catalytic processes, for instance, $H_2$ or CO oxidation[34–37]. Yet, the majority of previous in situ/*operando* studies of Pd-based methane oxidation catalysis have used spectroscopic techniques with only a limited spatial resolution (e.g., X-ray absorption spectroscopy and X-ray photoemission spectroscopy)[38,39]. Although those methods provide element-specific information about the oxidation state and local coordination environment, including either mostly bulk or (sub)surface sites when using XAS or XPS, respectively, this information is integral (i.e., averaged over micron-size specimen areas). Consequently, if active species comprise only a small fraction of the specimen, as is typically the case with industrial catalysts, their elucidation becomes challenging[40]. Furthermore, the coexistence of multiple phases complicates the search for structure–activity correlations. In this context, studies using *operando* TEM experiments can address these challenges by attaining a sufficient spatial resolution to link directly the nanoscale dynamics, typical for redox reactions with metal nanoparticle catalysts, to the catalytic performance (activity, selectivity, and stability)[34,36].

Recent *operando* TEM studies have indicated that Pd catalysts engage in oscillatory phase transformations at nanoscale (reshaping and particle splitting) under methane oxidation conditions[34]. The Pd and PdO phases co-exist and form phase boundaries within a single particle, consistent with earlier studies[12,21]. Previous reports also studied collective phase oscillations that are linked to oscillations of catalytic activity and investigated how the amplitude and frequency of the oscillations depend on gas composition and temperature[17,41]. While insightful, these findings are still insufficient to unambiguously identify the active surface state, the origin of phase oscillations and the influence of phase oscillations on catalytic activity. A deeper understanding of these key research questions can be achieved via temporally and spatially resolved atomic-level direct observation of the working state during methane oxidation conditions, including also the investigation of dynamic changes of Pd NPs as a function of the chemical potential of the gas phase. Herein, we utilize *operando* TEM, that is, real-time electron microscopy imaging coupled with online MS, complemented with surface studies using NAP-XPS to probe the active state, and with the aid of DFT calculations, to derive structure-performance relationships that govern methane oxidation on Pd NPs. We show how the size, phase composition and structural dynamics of Pd NPs respond to changes in the gas-phase chemical potential. We reveal the catalytically active state (phase coexistence and oscillations) and structures down to the atomic level and offer insights into the underlying reaction mechanism as well as the origin of phase oscillations under methane oxidation conditions.

## Results

### Oxidative treatment of Pd particles

Supplementary Fig. S1a shows a TEM image of the as-obtained Pd NPs (Sigma-Aldrich). The NPs feature an elongated shape with sizes ranging from 20 to 150 nm (Supplementary Fig. S1a). Selected-area electron diffraction (SAED) and high-resolution TEM (HRTEM) images reveal that the Pd particles are metallic with a *fcc* structure (Supplementary Fig. S1b, c). Before their use in methane oxidation, Pd NPs were pretreated using a Micro-Electro-Mechanical System (MEMS)-based in situ nanoreactor in 20% $O_2$ in He ($p(O_2)$ = 36 mbar) at 300 °C for 10 h to remove possible carbonaceous deposits and contaminants. Although the pretreated Pd NPs show no obvious morphological changes (Fig. 1a), SAED analysis reveals the coexistence of both Pd and PdO phases (Fig. 1b), suggesting that Pd is partially oxidized during the $O_2$ pretreatment. HRTEM imaging under 36 mbar $O_2$ reveals a core-shell microstructure of the calcined NPs, with a metallic core encapsulated by an oxide shell that is ca. 2–5 nm thin (Fig. 1c).

### Influence of temperature and gas-phase composition on particle dynamics, shape, and size

Starting from the calcined Pd NPs (Pd/PdO core/shell structures), we switched the gas phase from 20% $O_2$ in He ($p(O_2)$ = 36 mbar) to the reactive atmosphere containing 22% $CH_4$ and 4.9% $O_2$ ($p(CH_4)$ = 39.5 mbar, $p(O_2)$ = 8.8 mbar) in He at 350 °C ($p$(total) =180 mbar). In situ imaging shows that there are no obvious changes of the Pd particles both during the gas switching and after the stabilization of the gas phase. Next, we increased the temperature from 350 to 800 °C within 10 min (Fig. 1d–g and Supplementary Video 1) to study how the particle shape and size respond to the change in temperature. In situ observation during this temperature increase shows no changes in the calcined NPs up to 460 °C (Fig. 1d and Supplementary Fig. S2a, b). However, when the temperature is increased from 460 to 590 °C, hillocks of reduced Pd start to appear and grow on the surface of the particles (Supplementary Fig. S2b–f and Fig. 1e). This process continues and results in surface reconstruction and, eventually, fragmentation of particles. As the temperature increases further from 590 to 800 °C, the particles begin to sinter (Fig. 1f, g and Supplementary Fig. S2g–i), finally leading to the formation of metallic Pd at 800 °C (Supplementary Fig. S3). Interestingly, the particle sintering induced by high temperature is reversible, i.e., large metallic particles split into smaller particles when the temperature is decreased from 800 to 550 °C (Fig. 1h–k, Supplementary Fig. S4, and Supplementary Video 2). Having observed the effect of temperature, we further investigated how the gas-phase composition influences the size and shape of Pd particles. In situ TEM observation while adding $CH_4$ into the $O_2$/He flow at 550 °C reveals a gradual fragmentation of the particles (Fig. 1l–o and Supplementary Video 3). SAED study reveals an increased Pd to PdO ratio after the $CH_4$ addition (Supplementary Fig. S5). A similar fragmentation has been observed in the case of Cu particles in a redox atmosphere containing $O_2$ and $H_2$[34], and was explained by oxidation and subsequent reduction of particles that occur repeatedly due to the co-presence of both reducing and oxidizing species at a comparable chemical potential. Continuous in situ observation further shows that the particles do not split into ever smaller particles but rather their size stabilizes around a certain range (ca. 5–45 nm, Supplementary Fig. S6) under the conditions applied, indicating that the particle size is a function of temperature and gas composition.

To summarize, the in situ observations made during the heating and subsequent cooling, as well as during the gas switching, demonstrate directly that the particle size and structural dynamics of Pd are governed by the chemical potential of the gas phase. Since the particles of Pd adapt their shape, phase composition (Pd and PdO) and surface structure to the surrounding environment, the state of Pd observed ex situ, after passing through a temperature drop and change in atmosphere, does not represent the active state under reactive conditions.

### Identification of the phase composition and detection of catalytic activity

To gain insights into the active state of Pd NPs during methane oxidation and uncover the structure-performance relationships, we turned to experiments that combine in situ TEM observations with online MS analysis of the effluent gas (*operando* TEM). As shown in Fig. 2a, b and Supplementary Video 4, the particles display no obvious structural dynamics at 350 °C. This is evident from only a minor difference observed when comparing images taken at different times

(Fig. 2a, b), as shown in Fig. 2c by green contrast (this contrast was obtained by comparing Fig. 2a, b; a nearly black background of the whole image indicates minimal changes, see Supplementary Information for details). However, at 550 °C the particles indeed show a dynamic behavior (Fig. 2e, f and Supplementary Video 5) that leads to constant morphological changes and migration of particles (Fig. 2g, green contrast).

In situ electron diffraction was used to identify phases present under these reaction conditions. Analyses of the SAED patterns and the corresponding radial intensity profiles indicate that Pd and PdO co-exist both at 350 °C and 550 °C (Fig. 2d, h). The diffraction spots show almost no changes with time at 350 °C, implying no dynamic changes (Supplementary Fig. S7 and Supplementary Video 6). In contrast, the

diffraction spots are changing over time at 550 °C (Supplementary Fig. S8 and Supplementary Video 7), implying the presence of structural dynamics, in line with the TEM imaging. A comparison of PdO(101) to Pd(111) peak intensity ratio at 350 and 550 °C reveals a higher relative fraction of $Pd^0$ at 550 °C (Supplementary Fig. S9), consistent with the decreasing oxygen chemical potential with temperature[42,43]. In situ observation at medium magnifications further reveals that structural dynamics involve particle reshaping, sintering, outgrowth, and splitting (Fig. 2i–l and Supplementary Video 8). The presence of these dynamics is a consequence of the competing oxidizing and reducing processes near the Pd/PdO phase boundary. Note that the electron dose rates used for aforementioned in situ observations are considerably low, i.e., merely 250–700 $e \cdot nm^{-2} s^{-1}$. Under such

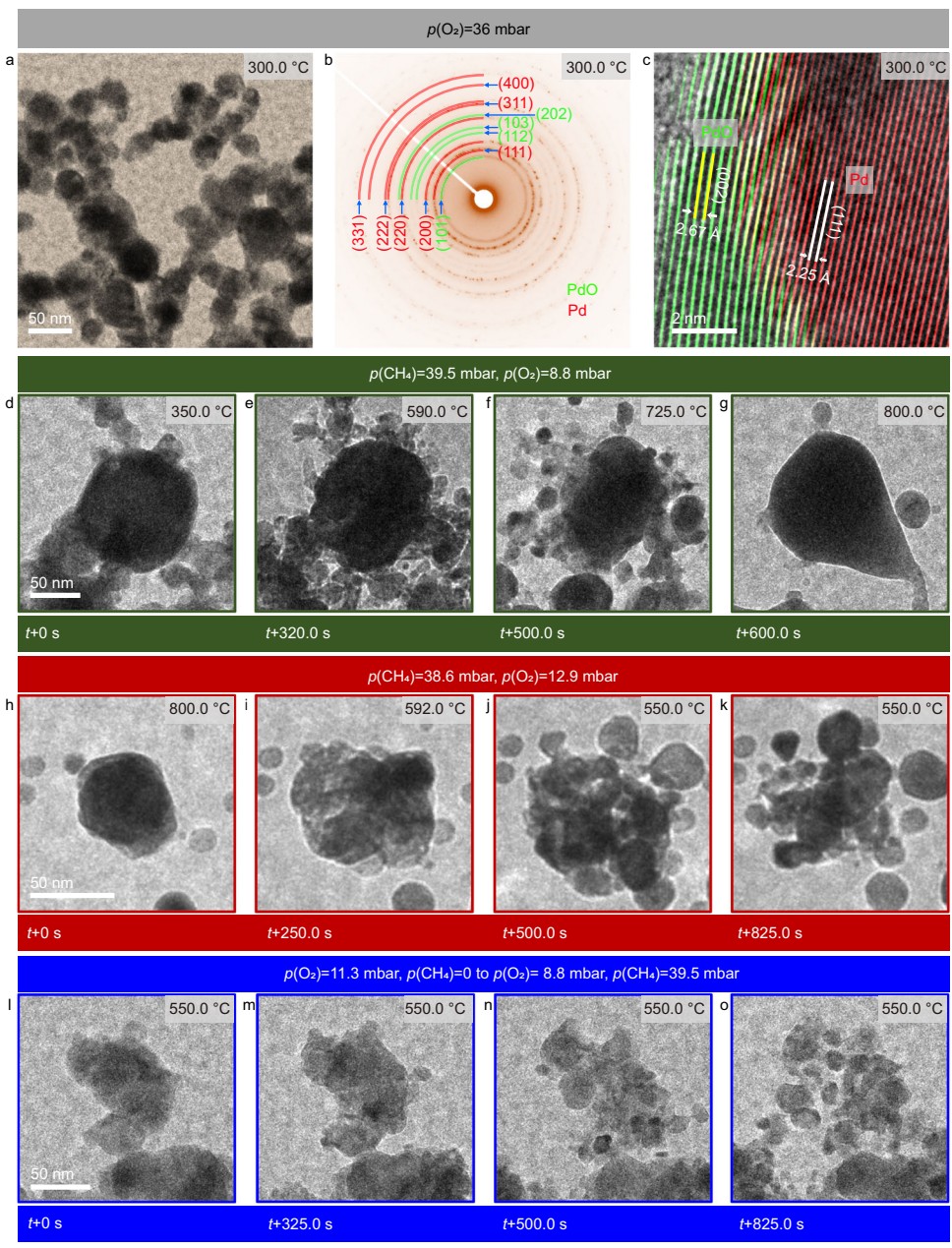

**Fig. 1 | Morphological and structural changes with varying temperatures and atmospheres.** TEM image, SAED patterns and HRTEM image of Pd NPs recorded after in situ calcination at 300 °C in 20% $O_2$ for 10 h (**a–c**). In situ observation of redox dynamics during increasing temperature from 350 to 800 °C ($p(CH_4)$ = 39.5 mbar, $p(O_2)$ = 8.8 mbar, and $p(He)$ = 131.7 mbar) (**d–g**). In situ observations of particle fragmentation during temperature decrease from 800 to 550 °C ($p(CH_4)$ = 38.6 mbar, $p(O_2)$ = 12.9 mbar, and $p(He)$ = 128.5 mbar) (**h–k**) and during the addition of $CH_4$ into the $O_2$/He flow ($p(CH_4)$ = 39.5 mbar, $p(O_2)$ = 8.8 mbar) (**l–o**). Electron dose rates for (**d–g**), (**h–k**), and (**l–o**) were about 460, 1110, and 710 $e \cdot nm^{-2} s^{-1}$, respectively.

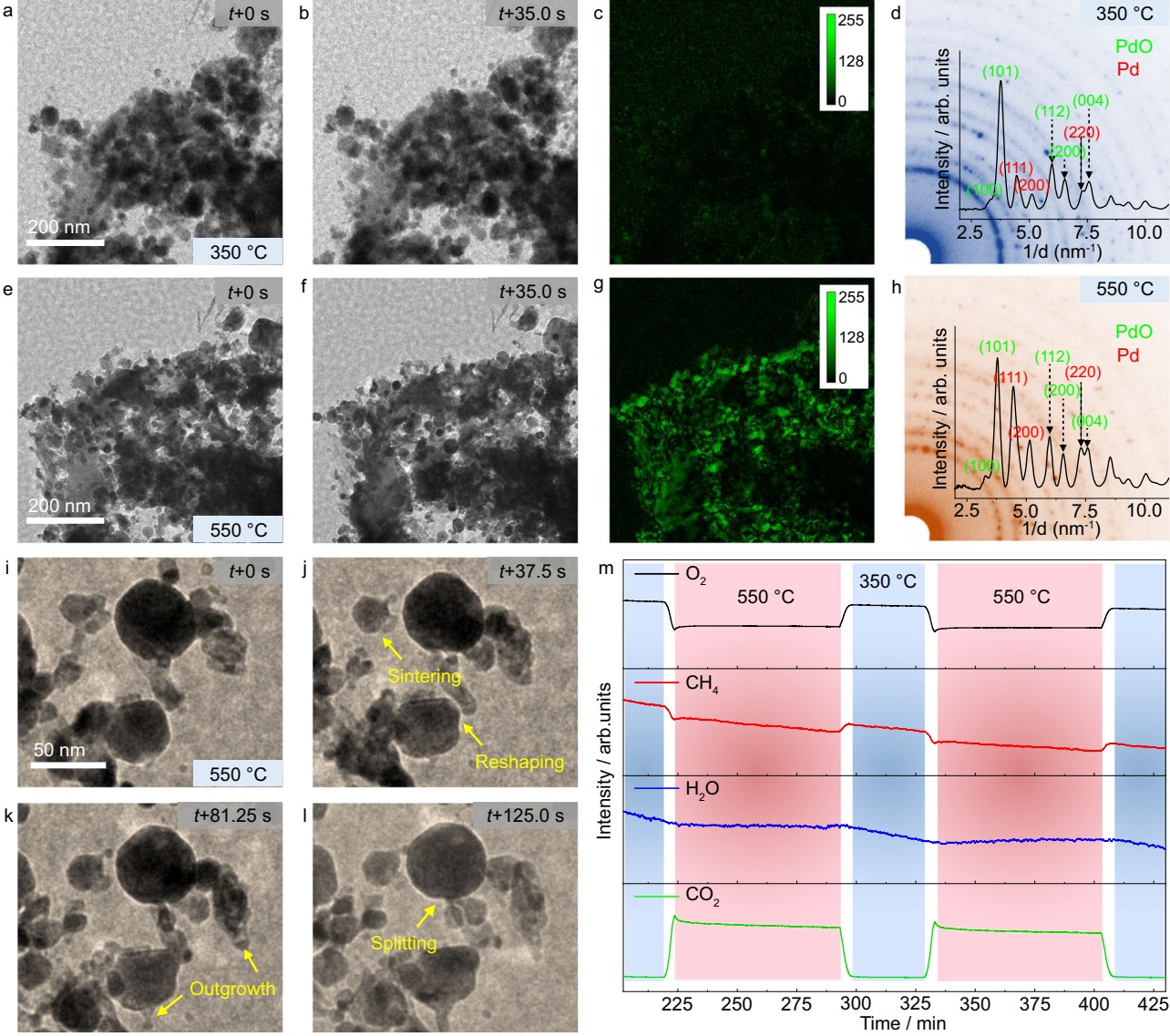

**Fig. 2 | Correlation between structural dynamics and catalytic activity.** In situ observation of the catalyst dynamics at 350 °C (**a**, **b**) and 550 °C (**e**, **f**) in the reactive atmosphere ($p(CH_4) = 39.5$ mbar, $p(O_2) = 8.8$ mbar and $p(He) = 131.7$ mbar). **c** Difference between images on (**a**, **b**). **g** Difference between images on (**e**, **f**). **d**, **h** Electron diffraction patterns recorded at 350 °C and 550 °C, respectively. (**i–l**) Particle dynamics observed at a medium magnification at 550 °C. **m** MS data recorded during *operando* TEM experiments. Electron dose rates for (**a**, **b**), (**e**, **f**), and (**i–l**) were 250, 700, and 700 e·nm$^{-2}$ s$^{-1}$, respectively.

applied dose rates, no influence of the electron beam could be detected. This is further supported by a control experiment in which the electron beam was cut off between shots to minimize the extent of electron irradiation. This experiment shows significant changes in the particle shape and relative location, similar to those shown in Fig. 2e, f and Fig. 2i–l, suggesting that the dynamics are not driven by the electron beam (Supplementary Fig. S10).

Turning now to analysis of the gas composition by a mass spectrometer connected to the outlet of the in situ TEM nanoreactor (Fig. 2m), the collected MS data shows a sharp increase in the $CO_2$ signal intensity and, simultaneously, a decrease of the $CH_4$ and $O_2$ signal intensity that coincide with the onset of redox dynamics at 350–550 °C. This data suggests clearly that the observed structural dynamics at 550 °C are linked to catalytic activity. The CO signal is also present in the MS data (Supplementary Fig. S11), yet the intensity of the detected CO signal is consistent with that expected from the fragmentation of $CO_2$ by the electron impact ionization method of the MS[44]. In other words, only $CO_2$ is produced in the catalytic reaction

even under the $O_2$-lean conditions used in this work, which agrees well with the previous studies that have been conducted under similar experimental conditions[41,45]. Yet, since the redox dynamics of individual NPs are mutually decoupled, MS data shows only an integral (averaged) signal and therefore oscillations reported in previous studies[41,45] are not seen in the MS data of this work.

### High-resolution observations of redox dynamics and interfacial structures

Having demonstrated the catalytic activity of Pd NPs and its correlation with structural dynamics, we performed in situ high-resolution observations to obtain atomic-scale information about the transient structures of the catalyst under reaction conditions. It should be mentioned that high-resolution imaging typically requires high electron dose rates that may induce beam effects, such as beam-induced reduction[46]. We compared the structural dynamics recorded under high and low dose rates, and found qualitatively similar results (see Fig. 3 and Supplementary Fig. S12), indicating that the applied dose

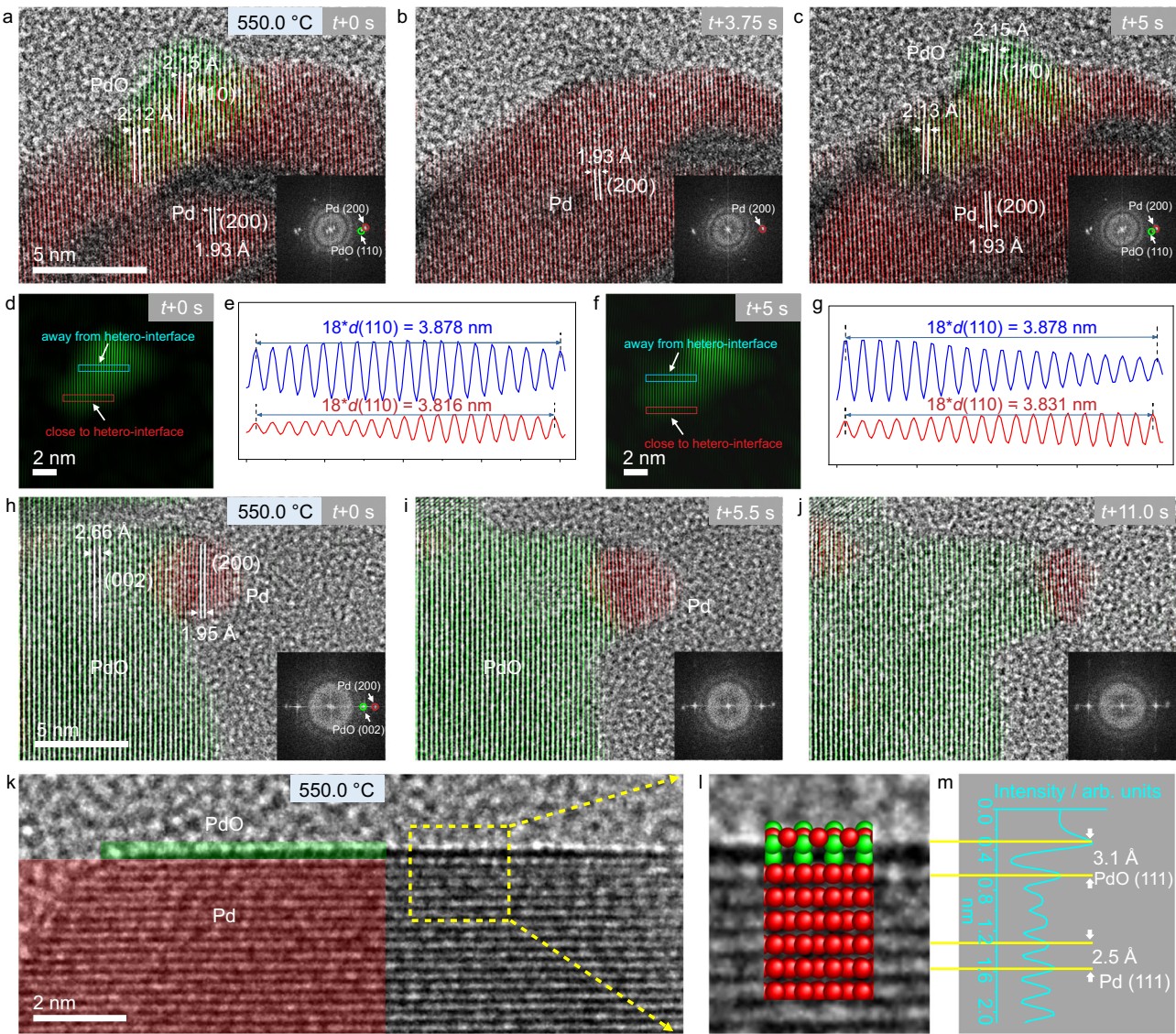

**Fig. 3 | In situ atomic-scale observations of Pd redox dynamics.** Oscillatory phase transition between Pd and PdO on the metallic surface (**a**–**c**) and the oxide surface (**h**–**j**) at 550 °C in the gas mixture of $CH_4$, $O_2$, and He ($p(CH_4) = 36.7$ mbar, $p(O_2) = 16.7$ mbar and $p(He) = 126.6$ mbar). Insets show FFTs of the corresponding HRTEM images. **d**, **e** Inverse Fourier transform image and line profiles of regions close to and away from the PdO/Pd interface. **f**, **g** Inverse Fourier transform image and line profiles of regions close to and away from the PdO/Pd interface. **k** HRTEM image and (**l**) enlarged HRTEM images from the dashed rectangle in (**k**), revealing the presence of a monolayer $PdO_x$ on Pd. The inset of (**l**) illustrates the atomic model. **m** Lattice $d$-spacing analysis. Electron dose rates for (**a**–**c**), (**h**–**j**), and (**k**) were $2.5 \times 10^5$, $1.6 \times 10^5$, and $2.5 \times 10^5$ e·nm$^{-2}$ s$^{-1}$, respectively.

rates for atomic-level imaging do not have a significant impact on the observed phenomena. Even though the electron irradiation might cause a change in the local chemical potential, it can be effectively compensated by varying either the partial pressure of gas phase or the temperature. In situ observations at atomic-scale not only confirm the presence of Pd and PdO phases and their oscillatory phase transition in individual particles (Fig. 3), but reveal further that the interconversion takes place on the surfaces of both, metal and metal oxide crystallites, as discussed in detail below.

Figure 3a–c shows snapshots from Supplementary Video 9 exhibiting oscillatory phase transition at a metallic surface. At 550 °C in the gas mixture of $CH_4$, $O_2$, and He ($p(CH_4) = 36.7$ mbar, $p(O_2) = 16.7$ mbar and $p(He) = 126.6$ mbar), we observe a periodic emergence and disappearance of a small oxide domain (green highlight) on the metal surface (red highlight). The transiently formed PdO domain shows semi-coherence with the underlying metallic particle, where PdO(110) is parallel to Pd(200). This phase epitaxy agrees with the previous

report on the oxidation (or reduction) of Pd in undiluted oxygen (or hydrogen) and CO oxidation[47–50]. Due to the lattice mismatch between PdO and Pd, a slight tilting (2.5–3°) of the PdO(110) to Pd(200) is observed, which evidences the interfacial strain. The presence of lattice strain between metal and metal oxide leads to a shrinkage of PdO(110) $d$-spacing at the interface region from 2.25 to 2.12–2.13 Å. The compressed lattice $d$-spacing is better visualized by the inverse Fourier transform image of PdO and the corresponding line profiles, as shown in Fig. 3d–g.

Figure 3h–j shows the structural dynamics occurring on the surface of an oxide particle (Supplementary Video 10). The supported Pd nanoparticle (red highlight) is reducing in size, due to the phase transition from Pd to PdO. A crystallographic relationship between the two phases is identified, where Pd(200) aligns parallel to PdO(002) (or PdO(101)). This relationship holds during the phase transition process, demonstrating that the phase transition is an order-to-order transition. Notably, the lattice bending is observed in

the vicinity of the metal oxide interface, suggesting the existence of the lattice strain.

In addition, the formation of a single-layer $PdO_x$ on Pd is also identified (Fig. 3k). Structural analysis reveals that the lattice distance between the topmost layer and the second one is about 3.1 Å, which is notably larger than lattice $d$-spacings of metallic Pd, suggesting that it is oxidic (Fig. 3l, m). The underlying Pd shows lattice fringes with a $d$-spacing of 2.25 Å, corresponding to (111) planes of $fcc$ structured Pd.

Overall, in situ high-resolution observations suggest that an active Pd catalyst is composed both of metal and metal oxide phases that interconvert dynamically under reaction conditions, in line with the results collected at a low magnification discussed above. In addition, the surfaces of NPs contain no obvious carbonaceous deposits, indicating that the catalyst is efficient in transforming $CH_4$ to $CO_2$. Associated with the oscillatory phase transition is the on-going formation of interfaces between metal and oxide domains. The strained coherent Pd/PdO interfaces might play a role in the catalytic process and will be discussed in more detail in the theoretical section of this work (vide infra).

## Surface composition and the electronic state of Pd NPs studied by NAP-XPS

To investigate the surface composition and the electronic state of Pd, NAP-XPS experiments were performed (experimental details are provided in the Supporting Information file). XPS data collected at 350 °C in 1 mbar $O_2$ (Fig. 4a) shows that the Pd $3d$ region contains peaks that can be fitted with two major components at binding energy (BE) of 334.9 eV and 336.1 eV, assigned to $Pd^0$ and $Pd^{2+}$ electronic states, respectively[51,52]. The $Pd^{2+}$ electronic state is likely due to the PdO phase observed in our TEM study described above. The incomplete oxidation of Pd to PdO (ca. 1:1 ratio of $Pd^0$ to $Pd^{2+}$ according to the fittings results shown in Supplementary Table S1) implies that under 1 mbar $O_2$ and at 350 °C the oxidation kinetics are slow (vide infra).

Increasing the temperature to 550 °C in 1 mbar $O_2$ leads to the evolution of peaks in the Pd $3d$ region such that only $Pd^{2+}$ peak remains, explained by the oxidation of $Pd^0$ and formation of PdO (Fig. 4b). The co-feeding of $CH_4$ to the $O_2$ flow ($CH_4$: 2.25 ml/min; $O_2$: 0.5 ml/min) at 550 °C leads to notable changes in the Pd $3d$ region (Fig. 4c). These changes are associated with the appearance of $Pd^0$ state and the disappearance of $Pd^{2+}$ state, explained by the reduction of PdO to metallic Pd. In addition, a feature with the intermediate (between $Pd^{2+}$ and $Pd^0$) BE energy appears at 335.3 eV, denoted $Pd^{\delta+}$ (dark-red line). A peak at this BE has been previously ascribed to a surface oxide $PdO_x$[52,53]. This assignment is in line with the in situ observation of the formation of a $PdO_x$ monolayer on the Pd metal seen in Fig. 3k. However, a partially reduced PdO surface may also contribute to the presence of this peak. Decreasing the temperature from 550 to 350 °C in the $CH_4/O_2$ mixture results in a notable decrease in the intensity of the $Pd^0$ peak, accompanied with the reappearance of the $Pd^{2+}$ peak along with an increase of the relative intensity of the $Pd^{\delta+}$ peak (Fig. 4d). Those results correlate with a higher chemical potential of oxygen at 350 °C relative to 550 °C. The subsequent increase of temperature from 350 to 550 °C, without changing the gas-phase composition, leads to the disappearance of the $Pd^{2+}$ peak and restores the ratio between $Pd^{\delta+}$ and $Pd^0$ peaks (1:4.4) seen prior to the initial lowering of the temperature from 550 °C to 350 °C, indicating a high reversibility of changes in the electronic states of Pd (Fig. 4c, e).

Turning now to the analysis of the O $1s$ region (Supplementary Fig. S13 and Supplementary Table S2), experiments with methane oxidation at 550 °C display peaks with a BE of 536.4 eV and 535.0 eV, ascribed to $H_2O$ and $CO_2$, respectively, due to the presence of these gases near the specimen surface[54–56]. Consistent with the MS data of the $operando$ TEM experiment, no XPS peak of gaseous CO is detected, suggesting the complete oxidation of $CH_4$. Notably, the peaks of the

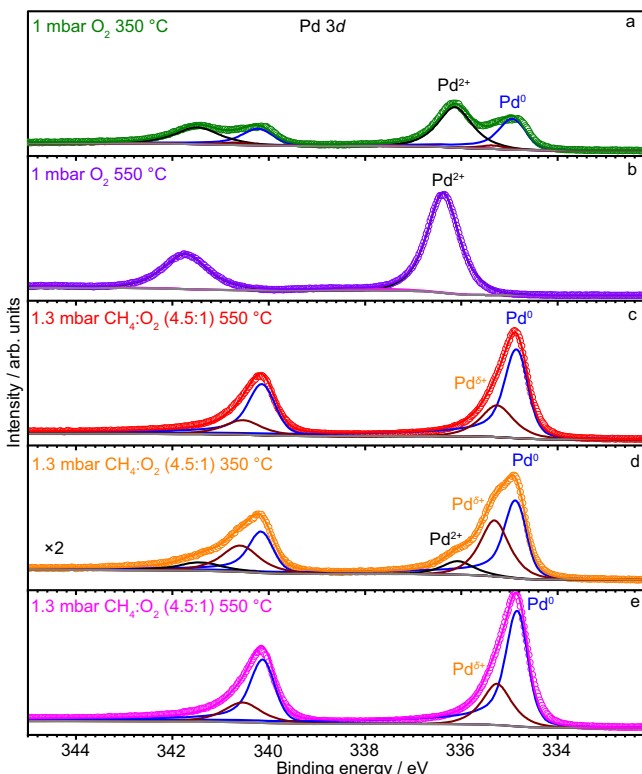

**Fig. 4 | In situ XPS under $O_2$ and the gas mixture of $CH_4$ and $O_2$.** Pd $3d$ XPS data of Pd NPs in **a** 1 mbar $O_2$ at 350 °C, **b** 1 mbar $O_2$ at 550 °C, **c** 1.3 mbar $CH_4/O_2$ = 4.5:1 at 550 °C, **d** 1.3 mbar $CH_4/O_2$ = 4.5:1 at 350 °C, and **e** 1.3 mbar $CH_4/O_2$ = 4.5:1 at 550 °C after temperature decrease.

gas-phase $CO_2$ and $H_2O$ are almost invisible in the O $1s$ XPS region when measured at 350 °C, confirming a lower catalytic activity at 350 °C as compared to 550 °C (Supplementary Fig. S13). Expectedly, when the temperature is increased from 350 to 550 °C, the contribution from methane oxidation products, $H_2O$ and $CO_2$, in the gas-phase increases (Supplementary Fig. S13), which correlates with the re-establishing of $Pd^{\delta+}$ and $Pd^0$ with the ratio of ca. 1:4.4 (Fig. 4e).

To summarize, NAP-XPS results show clearly that the surface chemical state of the Pd NP catalyst is highly sensitive to the gas-phase composition and temperature (Fig. 4 and Supplementary Fig. S13), underlying thereby the relevance of in situ characterization methods for the understanding of its active state. The formation of $CO_2$ and $H_2O$ at 550 °C and at a lower rate at 350 °C is consistent with the MS data collected during $operando$ TEM study, demonstrating that the catalyst is active in the dynamic state, displaying the $Pd^{\delta+}:Pd^0$ ratio of 1:4.4 while producing merely the full oxidation products. Additional discussion on XPS data and results (Supplementary Tables S3 and S4) based on first principles are provided in Supporting Information.

## DFT calculations

DFT calculations were carried out to understand the nature of the phase transitions and help identify structures active in methane oxidation. These simulations were performed using the Quantum ESPRESSO package[57] with the GGA-PBE exchange and correlation potential, full computational details are given in the supporting information. Following the experimental observations, we computed the stability of different Pd(100)/O surface terminations as a function of the gas-phase chemical potential by way of ab initio atomistic thermodynamics. This thermodynamic analysis shows that PdO is the stable phase up to 635 °C (or up to ca. 745 °C with entropic corrections for the solid[58]) at the experimental $O_2$ pressure, while the single-layer

PdO$_x$ is only metastable (Supplementary Fig. S14). Therefore, the experimental observation of metallic Pd and single-layer PdO$_x$ is driven by the kinetics of the methane oxidation reaction.

To gain insight into the kinetically driven phase transitions, we computed the reaction energetics of methane oxidation on different possible surfaces revealed by in situ observations, i.e., clean Pd, strained PdO, unstrained PdO, and Pd/PdO (i.e., a PdO monolayer on Pd shown in Fig. 5), see Supporting Information for details. While previous studies have focused typically only on the first C–H activation step of methane (owing to the high gas-phase barrier for this step requiring 2.5 eV[59]), we simulated all four C–H activation steps involved in methane oxidation to gain a better understanding of the complete reaction path. We found the reaction proceeds by the sequential transfer of H from the adsorbed CH$_4$ to oxygen on both the metal (adsorbed O*) and oxide (lattice oxygen sites), although other pathways may be possible on the pristine metal as O* blocks metal sites[60,61]. The adsorbed oxygen on the metal facilitates the C–H bond breaking, which is structure insensitive on the oxide[62]. On the metallic Pd(100) surface, we found the first and second C–H activation steps to be of similar energy, lower than that of the third and fourth C–H activation steps (Supplementary Table S5). Therefore, the first and second steps are predicted to be rate-limiting according to the universal Brønsted-Evans-Polyani (BEP) relationship[63–66] (Supplementary Fig. S15), i.e. the initial activation of methane is slow, as is expected by comparison to gas-phase energies. Estimating the barriers from BEP suggests an activation energy of about 1.0 eV for these steps (Supplementary Table S5)[67], which would make the metal surface an excellent catalyst when considering that the gas-phase barrier is 2.5 eV. However, dehydrogenation of methane on the metallic surface is predicted to be slow compared to surface oxidation through the dissociation of gas-phase O$_2$, where the barrier anticipated from BEP is 0.7 eV (Supplementary Table S5). Thus, the metallic surface phase is expected to oxidize toward the thermodynamically favored oxide phase or a metastable surface oxide.

On the unstrained PdO surface, the reaction mechanism is qualitatively different from that simulated on Pd(100) with O* since the C–O bonds form rapidly after both the first and second C–H activation steps on PdO[68]. That is, a C–O bond is formed after the first C–H activation of the adsorbed CH$_4$, resulting in OH and O–CH$_3$, the latter formed after the methyl group migration from Pd to O (see Fig. 5 and Supplementary Tables S6 and S7). Once the O–CH$_3$ species is formed, a second hydrogen atom is transferred to a second oxygen on the surface and the remaining O–CH$_2$ fragment is then further oxidized to OCH$_2$O species (Fig. 5). This OCH$_2$O fragment transfers a hydrogen atom to a surface oxygen to give a formate-like OCHO species. In the final C–H activation step, hydrogen is transferred from the formate to surface oxygen, yielding CO$_2$*. Our DFT calculations predict that the C–O bond formation significantly reduces the barriers associated with methane activation, which has two important consequences. The first consequence is that rather than the first step being rate-limiting, we find through the universal BEP relationship that the third dehydrogenation (i.e., hydrogen transfer from OCH$_2$O to surface oxygen) is rate-limiting on the oxide. We verified this BEP result by computing the activation energies by way of the climbing image nudged elastic band (CI-NEB) method, see Supplementary Information for details. As with the BEP result, the activation energy for the third C–H activation step was found to be rate-limiting on the oxide (Supplementary Table S8). On the unstrained oxide, the first two steps of hydrogen activation exhibit lower barriers (0.63 eV and 0.84 eV) compared to the subsequent two steps with significantly higher barriers (1.46 eV and 0.99 eV). The reason for the low barriers of the initial stages of methane oxidation lies in the formation of C–O bonds in the first two steps (Fig. 5). The second consequence of the low-barrier C–H bond dissociation followed by a facile C–O bond formation in both of the first two C–H activation steps relates to the absence of CO* species as a reaction intermediate, fully consistent with our MS data.

Because the computed pathway is associated with the surface reduction (transforming O* species to OH*), it implies that the oxide

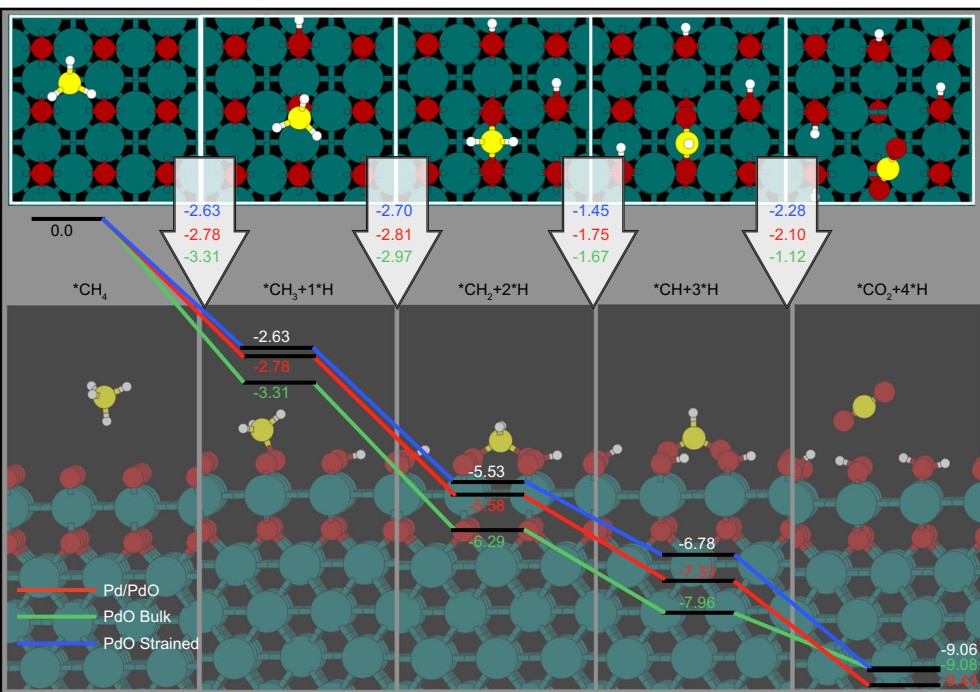

**Fig. 5 | Theoretical simulations.** Reaction energies for the complete methane oxidation on an unstrained bulk PdO(001) (green), strained PdO(001) (blue), and the strained Pd(100)/PdO(001) (red). The unstrained PdO(001) is constructed using the computed lattice parameter of bulk PdO, while the strained PdO(001) surface is constructed using the computed lattice parameter of bulk Pd metal, which is the same as that used to build the Pd(100)/PdO(001) surface. The green, red, yellow, and white spheres represent Pd, O, C, and H atoms, respectively, in the ball-and-stick model of the Pd(100)/PdO(001) structure shown in the background.

surface may get partially reduced during methane oxidation (C–H activation gives OH* groups that can condense to form O* and water that desorbs; repeating these steps leads to reduction of PdO). Overall, it is conceivable that reduction of PdO with methane as described above results in the formation of domains with metallic Pd. Due to the reoxidation by oxygen, oscillatory phase transitions may emerge since the barriers for reoxidation of the surface are competitive with surface reduction (Supplementary Fig. S16 and Supplementary Tables S6 and S7). Under such conditions, PdO is strained due to the lattice mismatch with metallic Pd (see Fig. 3). DFT calculations suggest that such strained PdO, whether multilayer (blue) or single-layer (red), will follow the same mechanism as unstrained PdO (green) but has more favorable energetics for methane oxidation. Straining the overlayer is found to reduce all the barriers, but the third C–H activation step remains rate-limiting with a computed activation energy of 1.12 eV, making the strained surface significantly more active than the unstrained surface (Supplementary Table S8). Moreover, reoxidation of the strained $PdO_x$ surface proceeds through $O_2$ dissociation with an activation energy of 1.11 eV. This observation suggests the oxidation and reduction rates may be similar, which gives rise to redox dynamics (Supplementary Fig. S17). Given that the phase coexistence has been observed in many redox-active metal catalysts, the kinetic hindrance of reoxidation of partially reduced oxide could be a general mechanism for the co-presence of both metal and metal oxide as well as their dynamic interconversion in redox reactions.

## Discussion

Real-time observations during temperature or gas-phase changes clearly reveal that the particle size, shape, and surface structures of Pd NPs are a function of the chemical potential of gas phase. Under the co-presence of $CH_4$ and $O_2$ at 550 °C, neither metallic Pd nor PdO is present as a static species, and a highly dynamic state characterized by phase coexistence and oscillatory transitions between Pd and PdO is observed. While phase coexistence and oscillations have been known from earlier studies[17,41,69], the real-time and space information of the associated dynamics are not well documented despite those might be the key to understand the catalytic function. In particular, both our *operando* TEM and these earlier studies demonstrate that such redox dynamics are correlated with the catalytic activity. The observation of the PdO → Pd phase transition points directly towards a Mars-Van Krevelen-like (MvK-like) mechanism, as the lattice O in PdO is consumed by methane[70]. Once the lattice O in PdO is depleted, it transforms into metallic Pd. Subsequently, the dissociative adsorption of $O_2$ on Pd leads to the reformation of PdO, through which the activity can be regenerated. Considering that PdO is the thermodynamically stable phase under these conditions, the presence of metallic Pd demonstrates the key role of reaction kinetics in determining the chemical state of the Pd catalysts.

High-resolution imaging further reveals the occurrence of oscillatory phase transition on the surface of both metal and oxide particles, with the transient formation of a strained and coherent interface between Pd and PdO (Fig. 3). Building on these atomic details, we have constructed models for use in first principles to understand their catalytic function. Ab initio simulations reveal that while PdO is thermodynamically stable, oscillatory phase transition occurs because the oxide is more effective in activating the C–H bonds of methane, which leads to surface reduction and ultimately may result in the reduction of oxide to metal (Supplementary Fig. S15 and Fig. 5). Conversely, the metal is ineffective at activating C–H bonds but easily activates $O_2$, causing oxidation of the metal. Thus, the preferential activation of reductant on the oxide and oxidant on the metal possibly induce the oscillatory phase transitions between the two states[71]. The appearance of strained PdO at the Pd/PdO interfaces during these phase oscillations can further enhance the C–H bond activation to improve the catalytic performance. Our

results thus suggest that one phase is not equally good at activating both reductant and oxidant in the gas phase, and an improved catalytic activity may be accessible if the system can be driven and stabilized at a dynamic state characterized by phase coexistence and cooperation[10,17,19]. In combination with our previous works on copper under different redox reactions[34,72,73], we can conclude that the emergence of catalytic activity is related to the dynamic interplay between coexisting phases, which generalizes it as an omnipresent mechanism for redox-active metal catalysts.

In summary, this work provides insights into the active structures of Pd catalysts and explains the origin of phase coexistence and oscillations in methane oxidation, which are of fundamental significance in deepening our understanding of Pd-based methane oxidation system and other metals-based redox catalytic systems. The dynamic picture of the constantly generated active sites/structures revealed by *operando* TEM, however, cannot be obtained by ex situ and *post mortem* characterizations or ensemble-average in situ techniques, and thus emphasizes the importance of *operando* TEM in uncovering the dynamic nature and catalytically relevant processes of catalysts. This work also highlights the importance of the complementary use of other in situ tools in catalysis research for advancing our understanding.

## Methods

### In situ/*operando* TEM experiments

The Pd NPs studied in methane oxidation were purchased from Sigma-Aldrich and were used as received. These Pd NPs, dispersed in chloroform, were drop-cast onto a MEMS-based heating chip. After drying in air, an oxygen plasma treatment was performed to remove organic residues. The heating chip was loaded into the DENSsolutions in situ gas-flow holder, after which the oxygen plasma treatment was repeated. The in situ holder was then inserted into the TEM chamber of an aberration-corrected JEM GrandARM 300 F transmission electron microscope and the inlet connected to the DENSsolutions gas-feeding system while the outlet gas manifold was connected to a quadrupole mass spectrometer (JEOL JMS-Q1500GC). After evacuating the gas-feeding system, 20% $O_2$ in He was flowed into the nanoreactor and the temperature was increased rapidly to 300 °C, where it was held for 10 h. Then the temperature was increased to 350 °C and $CH_4$ was added to the flow of $O_2$ in He to reach $p(CH_4) = 39.5$ mbar, $p(O_2) = 8.8$ mbar at $p$(total)=180 mbar. When the gas-phase composition stabilized, the temperature was increased from 350 to 800 °C in 10 min and then decreased to 550 °C in 5 min. The imaging was performed using Gatan OneView IS Camera at an acquisition rate of 4 frames per second with 2 K × 2 K pixels. The structural dynamics were recorded at 350 and 550 °C (at the aforementioned gas-phase composition) at different magnifications, while the composition of the off-gas flow was monitored by MS. In a separate experiment, $CH_4$ was added into the flow of $O_2$ in He at 550 °C ($p(CH_4) = 39.5$ mbar, $p(O_2) = 8.8$ mbar, $p$(total) = 180 mbar) and the structural changes of the Pd NPs were monitored. To study the temperature effect at this gas composition, the temperature was increased from 550 to 800 °C and then decreased to 550 °C ($p(CH_4) = 38.6$ mbar, $p(O_2) = 12.9$ mbar, $p$(total) = 180 mbar). The process was monitored using similar parameters as mentioned above. Atomic-scale observations were also carried out in situ to study the active structure of the catalyst at 550 °C ($p(CH_4) = 36.7$ mbar, $p(O_2) = 16.7$ mbar, $p$(total) = 180 mbar).

### NAP-XPS experiments

The NAP-XPS experiments were performed at the HIPPIE beamline of MAX IV Laboratory (Lund, Sweden). All measurements were performed in a catalytic cell placed at the solid-gas branch, the details of which are described elsewhere[74]. The powder samples were pressed into a 10-mm-diameter pellet. The samples were placed between a

stainless-steel sample holder and a lid (with a 6-mm-square hole). The samples were heated from the back side using an infrared laser and the temperature was measured by a chromel–alumel thermocouple spot-welded onto the sample plate. Photoelectrons from the sample are collected by a differentially pumped electrostatic lens system that refocuses the emitted electrons into the focal plane of a hemispherical electron energy analyzer. We recorded the Pd 3$d$ ($hv$ = 535 eV) and O 1$s$ ($hv$ = 740 eV) spectra with the same electron kinetic energy in order to obtain the same information depth of 5 Å in all experiments[75]. The overall spectral resolution was 0.1 eV in the O 1$s$ and 0.07 eV in the Pd 3$d$ regions. Binding energies at the core level (BE) were calibrated using the Fermi level. The accuracy of the BE calibration has been estimated to be around 0.1 eV.

All XPS spectra were recorded in normal photoemission mode. For quantitative XPS analysis, the least squares fit of the spectra was performed using the CasaXPS software (www.casaxps.com). The XPS line shape was assumed to be a Gaussian–Lorentzian function for the oxygen components and palladium oxide(s) and a Doniach–Sunjic function[76] for the Pd 3$d$ metallic component. A Shirley background was used to obtain the best fit.

### DFT calculations
All density functional theory (DFT) calculations were performed using the Quantum ESPRESSO package[57] using the GGA-PBE formalism[77] with projected augmented wave pseudopotentials[78] taken from the PSlibrary[79] with a kinetic energy (charge density) cutoff of 55 Ry (600 Ry). The surfaces were modeled using the PBE-optimized bulk palladium lattice parameter of 3.948 Å. A 3 × 3 supercell was used for all calculations with the periodic images separated by ~15 Å of vacuum to avoid spurious interactions. The bottom three layers of the slabs (six layers for the PdO surfaces) were fixed to their bulk positions and all other atoms were allowed to relax. Brillouin zone integrations were performed on a shifted 4 × 4 × 1 **k**-point mesh with Marzari-Vanderbilt cold smearing applied with a smearing value of 0.015 Ry[80]. The activation energies were computed by modeling the minimum energy paths (MEP) of the surface reaction using the climbing image nudged elastic band method (CI-NEB)[81,82]. The path for each NEB (Fig. 5 in the main text) was modeled as a separate MEP. Each MEP was modeled by taking ten images along the reaction pathway connecting the initial and final state and optimized until the force on the climbing image was below 0.05 eV/Ang. The activation energy is calculated as the energy difference between the transition state and the initial state along each MEP.

### Image processing
The reconstructed HRTEM images shown in Fig. 1c and Fig. 3a–c, h–j were obtained by overlapping the original HRTEM image with Fourier-filtered images of Pd and PdO fractions. The Fourier-filtered images of Pd and PdO can be obtained by performing a Fast Fourier Transform (FFT) of the original HRTEM image first and then masking the diffraction spots of Pd and PdO in the FFT image followed by an inverse FFT.

To compare images recorded after different time intervals and quantify the differences induced by structural dynamics, we first corrected the image drift using the "prealign Stack" plugin of ImageJ and then performed the "difference" function in the "Image Calculator" plugin of ImageJ. The intensity of each pixel in the newly generated image was calculated by img$_{new}$ = |img$_1$-img$_2$|. Therefore, if the difference between two compared images is large, the intensity given in the generated image is correspondingly high. At 350 °C, the Pd particles exhibit minimal changes over time, resulting in a small difference between Fig. 2a and b. Consequently, the resulting image predominantly features a nearly black background. However, at 550 °C, the Pd particles undergo dynamic changes in shape and location,

leading to a significant disparity between Fig. 2e and f. This substantial difference leads to a high contrast in Fig. 2f.

### Reporting summary
Further information on research design is available in the Nature Portfolio Reporting Summary linked to this article.

## Data availability
The data that support the findings of this study are included in the published article and its Supplementary Information files. These data are also available from the corresponding authors upon request.

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

## Acknowledgements

The authors acknowledge MAX IV Laboratory for time on Beamline HIPPIE under the 20230132 agreement. Research conducted at MAX IV, a Swedish national user facility, is supported by the Swedish Research Council under contract 2018-07152, the Swedish Governmental Agency for Innovation Systems under contract 2018-04969, and Formas under contract 2019-02496. Prof. C. Copéret from ETH Zurich is acknowledged for the support of this research. X.H. thanks the 1000 Talent Youth Project, Fuzhou University, Qingyuan Innovation Laboratory, and ETH Career Seed Grant SEED-14 18-2 for the financial support. C.S.P. would like to acknowledge DST-India for the INSPIRE Faculty Fellowship with Award number IFA-18 PH217 and the computing resources provided by Param Sanganak under the National Supercomputing Mission (NSM).

## Author contributions

X.H. and M.G.W. conceived the idea. X.H. carried out in situ / *operando* TEM experiments and analyzed the data. P.C.S. and T.J. conducted theoretical simulations. M.H. assisted with the MS measurement during in situ/*operando* TEM experiments. A.K. carried out the NAP-XPS experiment and data analysis. S.N.Y., Q.L., P.P.L., and W.Q.Y. analyzed the TEM data. A.F. and M.G.W. contributed to the valuable discussion. All authors participated in the writing and editing of the manuscript.

## Competing interests

The authors declare no competing interests.
