## [Peer Review File · Nature Communications]

Redox Dynamics and Surface Structures of an Active Palladium Catalyst during Methane OxidationREVIEWER COMMENTS

Reviewer #1 (Remarks to the Author):

The authors have submitted a manuscript on the behavior of Pd nanoparticles as catalysts for methane oxidation. The focus was on operando TEM and NAP-XPS combined with DFT calculations. The main novelty of the work are the operando TEM results that nicely show dynamic oscillations of Pd NP as response to different reaction conditions, i.e. temperatures and gas compositions. Although operando studies have been conducted in the past on Pd-based catalysts for methane oxidation, these studies have suffered from the fact that the obtained signals were averaged over the entire catalyst bulk (e.g. XAS methods) and this also holds for step-response experiments to reveal mechanistic details and reaction kinetics. In contrast, operando TEM applied in this study allowed to obtain spatially resolved data, which revealed the coexistence of oxidized PdO and Pd phases that are both involved in the reaction mechanism at the same time. On the contrary, the averaging techniques applied so far rather proposed partially oxidized surface PdOx as active phase - independent of static or dynamic test conditions have been applied. The coexistence of Pd and PdO phases during reaction, revealed by operando TEM, seems to be in agreement with DFT calculation and was supported by NAP-XPS. Based on these findings, the authors presented a more detailed picture of the nature of active Pd phases for methane oxidation, which is worth to be presented to the readership of Nature Communications.

Specific comments:

page 1: "Despite extensive studies, the nature of their catalytically active species and conceivable structural dynamics remains elusive."

This is a strong statement and I would recommend to soften it, since many details about the active sites and their dynamics have already been described in literature.

page 2: "However, while significant research efforts have been devoted to this catalytic system, our understanding of the working state of Pd catalysts is still insufficient for a rational development of improved catalysts."

I would skip the last part "for a rational development of improved catalysts", since this is a promise that catalysis research often makes, but that very often cannot be fulfilled, even if the active sites are known.

page 4: The sizes of the used Pd nanoparticles from 20 to 150 nm are rather big and above the sizes that are usually found in supported Pd catalysts.

page 10, bottom: "Pd(200)//PdO(002)" What does the "//" symbol mean? I guess that this could be unknown to part of the readership.

page 17, top: "On the Pd(100) surface, we found the first and second dehydrogenation steps of the adsorbed CH₄ to be of similar energy and rate-limiting..." This statement is not self-explaining. Why are the first two steps of dehydrogenation rate-limiting, particularly when the activation energies are unknown?

page 17, middle: "Rather than the first step being rate-limiting, we find the third dehydrogenation (i.e., hydrogen transfer from OCH₂O to surface oxygen) is rate limiting on the oxide."

See my comment above.

page 19: "combustion". I would avoid the term combustion and use oxidation instead for all flameless processes. This also holds for the Supporting Information.

Supporting Information

There are a number of typos that should be corrected.

Fig. S15: Most elementary steps are clear, but there are steps, in which water is desorbed resulting in a surface with an excess oxygen that cannot be explained from the shown reaction stoichiometry.

Reviewer #2 (Remarks to the Author):

This work studied the redox dynamics and surface structures of an active palladium catalyst during methane oxidation by using both experiment and calculation technology, in which it was shown the acylative state that is characterized by phase coexistence and oscillatory phase. This work is important for the people to understand the dynamics behavior of active site of Pd catalyst during the methane oxidation reaction condition, but there are some important points need to be addressed well before its acceptance.

Major comments:

(1) This work only gives the reaction energy change is not enough. For example, in Fig.S15, $\text{Pd} + \text{O} + \text{CH}_2 \rightarrow \text{Pd} + \text{OH} + \text{H}$, I am not sure it favors the path of CH_2 direct dissociation in the kinetic respect. Moreover, the active phase reported in the present work is controlled by kinetic of methane oxidation reaction, so study the kinetic path of methane oxidation by using DFT is necessary and important.

(2) Importantly, the theoretical part of this work is a qualitative result, and use more quantitative technology like kinetic Monte Carlo (kMC) to simulate phase transition at the reaction condition is a best choice. In fact, submit work to NC, a high-level journal, strong work should be done.

(3) For the role of O^* , either to active C-H bond, or to help C-O bond formation over Pd, should be revealed clearer. For examples, J. Chem. Phys. 2009, 131, No. 144704; J. Catal. 2011, 282, 74–82 ; Surf. Sci. 2016, 650, 210–220;...

(4) Some key steps maybe ignored in PdO, which can be refereed to J. Phys. Chem. C 2022, 126, 14201–14210, J. Am. Chem. Soc. 2015, 137, 12035–12044 and other works.

Others:

(5) Abstract: It is too general and should provide more substantive content.

(6) I cannot find the detail structure of Pd_xO_y overlayer, which is important in this work.

Reviewer #3 (Remarks to the Author):

The paper by Yue and colleagues entitled Redox Dynamics and Surface Structures of an Active Palladium Catalyst during Methane Oxidation reports on primarily in situ and operando TEM studies of Pd nanoparticles used in methane (slip) combustion. The observations made using TEM have then been underpinned by in situ XPS measurements and DFT simulations using Quantum Espresso. The key findings of the study demonstrate that in particular a Pd metallic nanoparticle with a PdOx surface seems to be particularly active at methane combustion rather than a Pd or PdO nanoparticle. These results are consistent with past studies on bulk samples but here the TEM and XPS identify a PdOx type species rather than a distinct PdO overlayer on the Pd metal. Additional insight into the behaviour of the Pd particles has also been shown – in particular the melting and fusing of Pd metal nanoparticles at high temperatures followed by fragmentation at lower temperatures.

I really enjoyed reading this paper. It is very well written and didactic. The images are striking and I think it remarkable that such data could be obtained under operando conditions and that the MS data could be recorded. I think the work should be published largely as is although I have two minor comments:

1) Figure S6 I find very interesting although I wonder if the authors could produce similar histograms

for figure 1 (g) @ 800 C and (k) @ 550 C to see if all Pd nanoparticles sinter/disperse or whether there are certain nanoparticles that are more susceptible to this?
2) Figure 2 C is difficult to see and interpret – could the contrast be improved?

Revised Manuscript Title: Redox Dynamics and Surface Structures of an Active Palladium Catalyst during Methane Oxidation

Manuscript ID: NCOMMS-23-43085-T

Dear Reviewers,

We are very grateful for your valuable feedback and comments, which have aided us greatly in the preparation of an improved manuscript. Please find below our point-by-point response to each comment of the reviewers. While the reviewers' comments are in regular text, our responses are in blue. Modifications of the main text are highlighted yellow. We hope that our revised Manuscript can be accepted for publication in *Nature Communications*.

Sincerely,

Dr. Xing Huang

Reviewer #1 (Remarks to the Author):

The authors have submitted a manuscript on the behavior of Pd nanoparticles as catalysts for methane oxidation. The focus was on operando TEM and NAP-XPS combined with DFT calculations. The main novelty of the work are the operando TEM results that nicely show dynamic oscillations of Pd NP as response to different reaction conditions, i.e. temperatures and gas compositions. Although operando studies have been conducted in the past on Pd-based catalysts for methane oxidation, these studies have suffered from the fact that the obtained signals were averaged over the entire catalyst bulk (e.g. XAS methods) and this also holds for step-response experiments to reveal mechanistic details and reaction kinetics. In contrast, operando TEM applied in this study allowed to obtain spatially resolved data, which revealed the coexistence of oxidized PdO and Pd phases that are both involved in the reaction mechanism at the same time. On the contrary, the averaging techniques applied so far rather proposed partially oxidized surface PdOx as active phase - independent of static or dynamic test conditions have been applied. The coexistence of Pd and PdO phases during reaction, revealed by operando TEM, seems to be in agreement with DFT calculation and was supported by NAP-XPS. Based on these findings, the authors presented a more detailed picture of the nature of active Pd phases for methane oxidation, which is worth to be presented to the readership of Nature Communications.

We thank Reviewer 1 for the generally positive evaluation of our study. Specific comments are answered below.

Q1: page 1: "Despite extensive studies, the nature of their catalytically active species and conceivable structural dynamics remains elusive." This is a strong statement and I would recommend to soften it, since many details about the active sites and their dynamics have already been described in literature.

A1: We have modified the text to now read as follows:

"Despite extensive studies and notable advances, the nature of their catalytically active species and conceivable structural dynamics remains only partially understood."

Q2: page 2: "However, while significant research efforts have been devoted to this catalytic system, our understanding of the working state of Pd catalysts is still insufficient for a rational development

of improved catalysts." I would skip the last part "for a rational development of improved catalysts", since this is a promise that catalysis research often makes, but that very often cannot be fulfilled, even if the active sites are known.

A2: We have removed the last part of the sentence, as suggested by Reviewer 1.

Q3: page 4: The sizes of the used Pd nanoparticles from 20 to 150 nm are rather big and above the sizes that are usually found in supported Pd catalysts.

A3: The initial Pd particles are indeed larger than those typically found in the supported Pd catalysts. However, our work demonstrates that the size of the Pd particles depends on the chemical potential of the gas phase, i.e., the size changes dynamically with changes in temperature and gas composition. When subjected to methane oxidation conditions, the Pd particles undergo fragmentation at 550 °C. After dynamic equilibrium is established, the average particle size of Pd decreases to 10 nm, which is comparable to the particle size in Pd catalysts reported previously.

Q4: page 10, bottom: "Pd(200)//PdO(002)" What does the "//" symbol mean? I guess that this could be unknown to part of the readership.

A4: The notation "Pd(200)//PdO(002)" indicates that the (200) plane of metallic Pd is parallel to the (002) plane of PdO. We acknowledge that some readers may not be familiar with this notation, thereby we have modified it as follows:

"A crystallographic relationship between the two phases is identified, where Pd(200) aligns parallel to PdO(002) (or PdO(101))."

Q5: page 17, top: "On the Pd(100) surface, we found the first and second dehydrogenation steps of the adsorbed CH₄ to be of similar energy and rate-limiting..." This statement is not self-explaining. Why are the first two steps of dehydrogenation rate-limiting, particularly when the activation energies are unknown?

A5: We acknowledge this comment of Reviewer 1. The text was revised to specify that this conclusion was made based on the Brønsted-Evans-Polyani (BEP) relationship:

"On the metallic Pd(100) surface, we found the first and second C–H activation steps to be of similar energy, lower than that of the third and fourth C–H activation steps. Therefore, the first and second steps are predicted to be rate-limiting according to the universal Brønsted-Evans-Polyani (BEP) relationship, i.e. the initial activation of methane is slow, as is expected by comparison to gas-phase energies."

Q6: page 17, middle: "Rather than the first step being rate-limiting, we find the third dehydrogenation (i.e., hydrogen transfer from OCH₂O to surface oxygen) is rate limiting on the oxide." See my comment above.

A6: For clarification, we have added results of the climbing image nudged elastic band (CI-NEB) method to Table S8. These results show that the step 3 has a higher activation energy over all oxide structures considered, and hence is rate limiting.

Q7: page 19: "combustion". I would avoid the term combustion and use oxidation instead for all flameless processes. This also holds for the Supporting Information.

A7: We have changed the term "combustion" to "oxidation" throughout the manuscript and SI.

Q8: There are a number of typos that should be corrected.

A8: We have reviewed the SI file and corrected any typos found.

Q9: Fig. S15: Most elementary steps are clear, but there are steps, in which water is desorbed resulting in a surface with an excess oxygen that cannot be explained from the shown reaction stoichiometry.

A9: The comment is acknowledged. Note that the text accompanying Fig. S15 discussed those steps as follows:

“Once the water molecule was created, it was assumed to desorb and was replaced by another oxygen atom to allow the reaction to progress in this low oxygen coverage regime.”

Reviewer 1 points out correctly that our original schematic in Fig. S15 did not show that O* species should be provided (2 times) to maintain the reaction stoichiometry. Our revised Fig. S15 now shows the reaction stoichiometry correctly.

Reviewer #2 (Remarks to the Author):

This work studied the redox dynamics and surface structures of an active palladium catalyst during methane oxidation by using both experiment and calculation technology, in which it was shown the acylative state that is characterized by phase coexistence and oscillatory phase. This work is important for the people to understand the dynamics behavior of active site of Pd catalyst during the methane oxidation reaction condition, but there are some important points need to be addressed well before its acceptance.

We thank Reviewer 2 for recognizing our work as important. In what follows, we address critical comments of Reviewer 2.

Q10: This work only gives the reaction energy change is not enough. For example, in Fig.S15, $\text{Pd}+\text{O}+\text{CH}_2\rightleftharpoons\text{Pd}+\text{OH}+\text{H}$, I am not sure it favors the path of CH₂ direct dissociation in the kinetic respect. Moreover, the active phase reported in the present work is controlled by kinetic of methane oxidation reaction, so study the kinetic path of methane oxidation by using DFT is necessary and important.

A10: We thank Reviewer 2 for this important comment. We agree that reaction kinetics is critical for the present system, and we have therefore used the universal Brønsted-Evans-Polanyi relationship to estimate the involved kinetic barriers. Following the suggestion of the referee, we have performed CI-NEB analysis to verify the BEP results and quantify the activation energies. We find that the BEP estimates of the barriers are qualitatively correct and now include both BEP and CI-NEB results on the activation barriers (see Table S8).

Q11: Importantly, the theoretical part of this work is a qualitative result, and use more quantitative technology like kinetic Monte Carlo (kMC) to simulate phase transition at the reaction condition is a best choice. In fact, submit work to NC, a high-level journal, strong work should be done.

A11: As discussed in A10, we have included quantitative results from the CI-NEB simulations to complement the qualitative BEP results. In addition to that, in the revised version of the manuscript, we have also included a simple microkinetic model to demonstrate how phase oscillations can emerge, presented below in Figure S17. We refer to the revised SI for details.

Figure S17. Oxide coverage vs time found using the DFT parameterized microkinetic model under 29.5 mbar CH₄ and 8.8 mbar O₂ at 550 C.

Q12: For the role of O*, either to active C-H bond, or to help C-O bond formation over Pd, should be revealed clearer. For examples, J. Chem. Phys. 2009, 131, No. 144704; J. Catal. 2011, 282, 74–82; Surf. Sci. 2016, 650, 210–220;...

A12: As discussed in our manuscript, our DFT model considers the involvement of O* species in the C-H bond activation (and hence H transfer). The choice of this model allows to compare O* species on the metal to lattice oxygen sites in the oxide. Furthermore, experimental TEM results show that even at high temperatures, the surface of Pd is covered by an oxide-like layer. Unlike on the oxide, the formation of a C–O bond on the metal surface proceeds only after the complete dehydrogenation. We have cited the requested references in the revised version as detailed below: “The adsorbed oxygen on the metal facilitates the C–H bond breaking, which is structure insensitive on the oxide. [J. Chem. Phys. 2009, 131, 144704]”.

To support our argument on the BEP relationships for bare metal surfaces, we have cited J. Catal. 2011, 282, 74–82.

Q13: Some key steps maybe ignored in PdO, which can be referred to J. Phys. Chem. C 2022, 126, 14201–14210, J. Am. Chem. Soc. 2015, 137, 12035–12044 and other works.

A13: We have incorporated the requested references into the revised text as follows:

“We found the reaction proceeds by a sequential H-transfer from the adsorbed CH₄ to adsorbed O on both the metal and oxide, although other pathways may be possible [J. Phys. Chem. C 2022, 126, 14201–14210, J. Am. Chem. Soc. 2015, 137, 12035–12044].”

Q14: Abstract: It is too general and should provide more substantive content.

A14: We have rewritten the abstract and enriched it with experimental results. We refer to the revised manuscript text for further details.

Q15: I cannot find the detail structure of Pd_xO_y overlayer, which is important in this work.

A15: Our data shows the lattice distance between the topmost layer and the second layer (about 3.1 Å) is larger than the lattice d-spacing of Pd 111 (2.25 Å), suggesting the oxidic nature of the

surface layer. However, due to the parallel orientation of this layer to the electron beam, we can only capture the cross-sectional view of this layer. Furthermore, the presence of thick SiN windows (about 60-80 nm), hinders the image quality of this overlayer, making it unsuitable for image simulation. These factors collectively pose challenges in conducting a detailed analysis and determining the structure of this layer. We believe that a windowless method combined with a surface sensitive technique within an AC-TEM machine (e.g., a second electron detector) could potentially provide atomic information about the structure of this overlayer. We intend to pursue such a study once the methodology becomes available to us.

Reviewer #3 (Remarks to the Author):

The paper by Yue and colleagues entitled Redox Dynamics and Surface Structures of an Active Palladium Catalyst during Methane Oxidation reports on primarily in situ and operando TEM studies of Pd nanoparticles used in methane (slip) combustion. The observations made using TEM have then been underpinned by in situ XPS measurements and DFT simulations using Quantum Espresso. The key findings of the study demonstrate that in particular a Pd metallic nanoparticle with a PdOx surface seems to be particularly active at methane combustion rather than a Pd or PdO nanoparticle. These results are consistent with past studies on bulk samples but here the TEM and XPS identify a PdOx type species rather than a distinct PdO overlayer on the Pd metal. Additional insight into the behaviour of the Pd particles has also been shown – in particular the melting and fusing of Pd metal nanoparticles at high temperatures followed by fragmentation at lower temperatures.

I really enjoyed reading this paper. It is very well written and didactic. The images are striking and I think it remarkable that such data could be obtained under operando conditions and that the MS data could be recorded. I think the work should be published largely as is although I have two minor comments:

We appreciate the highly supportive and encouraging feedback by Reviewer 3. We address two minor comments of Reviewer 3 below.

Q16: Figure S6 I find very interesting although I wonder if the authors could produce similar histograms for figure 1 (g) @ 800 C and (k) @ 550 C to see if all Pd nanoparticles sinter/disperse or whether there are certain nanoparticles that are more susceptible to this?

A16: *In situ* observations during the temperature increase from 350 °C to 800 °C revealed that Pd particles underwent fragmentation around 590 °C and subsequent sintering during the temperature increase from 590 to 800 °C. While the majority of particles sintered into larger ones (leading to a significant decrease in particle number), some particles only underwent a shape reconstruction and were resistant to sintering. At present, it is not clear what effect or structural feature distinguishes particles that undergo sintering from particles that undergo reconstruction upon the temperature increase.

Upon decreasing the temperature from 800 °C to 550 °C, particle fragmentation occurs, observed primarily in larger particles, while the smaller ones (ca. 10-20 nm) do not fragment further. The larger particles formed at 800 °C contain mainly metallic Pd, which is unstable at medium temperatures (550 °C) where the chemical potential of CH₄ and O₂ is comparable. So larger particles undergo repeated oxidation and subsequent reduction dynamics, leading to fragmentation. Smaller particles are already within the size range of particles that are dynamically

stable at 550 °C, thus no further fragmentation is observed. The particle size distributions at 800 °C and 550 °C are provided in Figure S3 and Figure S4 in the revised supplementary information.

Q17: Figure 2 C is difficult to see and interpret – could the contrast be improved?

A17: Figure 2c illustrates the disparity between Figure 2a and Figure 2b. By subtracting Image b from Image a, we can obtain a quantitative difference between the two images, *i.e.*, $\text{Image } c = | \text{Image } a - \text{Image } b |$. The greater the contrast in the final image, the more pronounced dissimilarity it represents. At 350°C, the Pd particles exhibit minimal changes over time, resulting in a small difference between Images a and b. Consequently, the resulting image predominantly features a nearly black background. However, at 550 °C, the Pd particles undergo dynamic changes in shape and location, leading to a significant disparity between Figure 2d and Figure 2e. This substantial difference leads to a high contrast in Figure 2f. We have provided a more detailed explanation on the observed image contrast in Supporting Information.

REVIEWERS' COMMENTS

Reviewer #1 (Remarks to the Author):

The authors responded satisfactorily to the reviewers' comments and have answered all open questions and comments.

Reviewer #2 (Remarks to the Author):

Authors have revised their work based on my comments, and the quality of this work was improved in the present from.

(1) What are the conditions for such kind of Redox Dynamics behavior to occur? Contains both oxides and reducers like O₂ and CH₄? If so, how about for Cu(Ru)-catalyzed propylene epoxidation (C₃H₆+O₂), co-existence of Cu(Ru) and Cu₂O(RuO₂)?

(2) Gives the detail information how to draw the Fig.S14 is necessary. For examples, the calculated DFT data, calculation model for Pd/Oads..

Reviewer #3 (Remarks to the Author):

I can see that the authors have gone to great lengths to address the points made in the first round of reviewing and as a result have produced a version of the manuscript which is more suitable for publication. I feel this version could be published as is although i note the caption for Figure 2 makes reference to images 3a and 3b as well as 3d and 3e when i think they mean 2a and 2b etc...

Revised Manuscript Title: Redox Dynamics and Surface Structures of an Active Palladium Catalyst during Methane Oxidation

Manuscript ID: NCOMMS-23-43085A

Dear Editor,

Please find below our response to the remaining comment of the reviewers. While the reviewers' comments are in regular text, our responses are in blue. Modifications of the main text are highlighted yellow. We hope that our revised Manuscript can be accepted for publication in *Nature Communications*.

Sincerely,

Dr. Xing Huang on behalf of all authors

Reviewer #1 (Remarks to the Author):

The authors responded satisfactorily to the reviewers' comments and have answered all open questions and comments.

We once again thank Reviewer 1 for constructive comments and suggestions.

Reviewer #2 (Remarks to the Author):

Authors have revised their work based on my comments, and the quality of this work was improved in the present form.

We thank Reviewer 2 for acknowledging that our manuscript has been improved. We answer below the remaining questions of Reviewer 2.

Q1. What are the conditions for such kind of Redox Dynamics behavior to occur? Contains both oxides and reducers like O₂ and CH₄? If so, how about for Cu(Ru)-catalyzed propylene epoxidation (C₃H₆+O₂), co-existence of Cu(Ru) and Cu₂O(RuO₂)?

A1. The key for triggering the redox dynamics is the comparable chemical potential for the transformation between the metal and oxide phases in conditions of the co-presence of reducing and oxidizing gases, within a specific temperature range and gas composition. For instance, in the present work, the co-existence of CH₄ and O₂ with a ratio of ca. 4.5:1 to 3:1 leads to redox dynamics through the oscillatory phase transitions between Pd and PdO at temperatures between 550 °C and 650 °C. Similarly, on Cu, the co-presence of H₂ and O₂ with a ratio of about 10:1 has been shown to lead to the redox dynamics and oscillatory phase transitions between Cu and Cu₂O at temperatures between 400 °C to 650 °C (see *Adv. Mater.* **2021**, *33*, 2101772). The dynamic behavior in the Cu system was also observed with an H₂ to O₂ ratio of 5:1, albeit at higher temperatures. In addition, our unpublished *operando* TEM study of C₂H₄ oxidation on Cu revealed redox dynamics and phase transitions (Cu ↔ Cu₂O) under an O₂ lean condition (note that the latter results are beyond the scope of the present study and will be reported separately).

Q2. Gives the detail information how to draw the Fig.S14 is necessary. For examples, the calculated DFT data, calculation model for Pd/O_{ads}.

A2. We have added PdO(001)/Pd, PdO(101)/Pd and Pd/O_{ads} structural models on the right panel of Figure S14 and modified the caption to include the adsorption energies and heat of formation of

PdO for clarity. The new figure and its caption are reproduced below:

Figure S14. Surface phase diagram computed by employing *ab initio* thermodynamics. Only the bulk oxide PdO, metal with adsorbed oxygen (less than 1/4 ML), and oxygen-free Pd appear as stable phases, as labelled. Yellow lines indicate the areas where the PdO(001)/Pd and PdO(101)/Pd interfaces are stable. The adsorption energies of O adatom per area used for plotting the phase diagram are -0.011 , -0.038 and -0.049 eV ($\text{O}^* \times \text{\AA}^2$)⁻¹ for PdO(001)/Pd, PdO(101)/Pd and Pd/O_{ads} respectively. The heat of formation of PdO was taken as -1.17 eV. The right panel shows the top view of PdO(001)/Pd, PdO(101)/Pd and Pd/O_{ads} structures.

Reviewer #3 (Remarks to the Author):

I can see that the authors have gone to great lengths to address the points made in the first round of reviewing and as a result have produced a version of the manuscript which is more suitable for publication. I feel this version could be published as is although i note the caption for Figure 2 makes reference to images 3a and 3b as well as 3d and 3e when i think they mean 2a and 2b etc... We thank Reviewer 3 for the recommendation to publish our work. The typos in the caption of Figure 2 were corrected in the revised manuscript.